# Un algoritmo GRASP para el sistema de remanufactura en tres etapas

**Teodor A. Diaconescu, Alberto Herrán, J. Manuel Colmenar**
Dpto. de Informática y Estadística, Universidad Rey Juan Carlos
C/. Tulipán, s/n, Móstoles, 28933 (Madrid), Spain
teodor.diaconescu@urjc.es
alberto.herran@urjc.es
josemanuel.colmenar@urjc.es

## Abstract

Este trabajo aborda el problema de la optimización de sistemas de remanufactura en tres etapas (3T-RSSP), cuyo objetivo es minimizar el tiempo de finalización de los trabajos del sistema (*makespan*). Este tipo de problemas se sitúa en línea con los Objetivos de Desarrollo Sostenible 9 y 12. Para resolver el problema se propone una metaheurística GRASP compuesta por un método constructivo y una búsqueda local que operan en las tres etapas de las instancias del problema. Los resultados experimentales, obtenidos sobre un conjunto de 34 instancias del estado del arte, demuestran que la propuesta GRASP logra un excelente equilibrio entre la calidad de la solución y el tiempo de cómputo, obteniendo el mejor valor promedio para la función objetivo empleando un tiempo de ejecución promedio muy pequeño en comparación con los métodos del estado del arte.

## 1 Introducción

El avance tecnológico y el crecimiento del consumo en las últimas décadas ha traído consigo un creciente aumento en el número de productos al final de su vida útil, conocidos como productos EOL (del inglés, *End-Of-Life*). Los dispositivos electrónicos, vehículos o componentes industriales son algunos de estos productos, cuyo volumen no para de crecer, generando problemas en el medio ambiente, la economía y la salud. Ante esta problemática, reconocida globalmente en la Agenda 2030 para el Desarrollo Sostenible de las Naciones Unidas [6], los modelos de economía circular han ganado protagonismo, mitigando estos desafíos para construir un futuro más sostenible [4]. Dentro de estos modelos se encuentran los sistemas de remanufactura. Estos sistemas llevan a cabo diferentes procesos para dar mantenimiento y/o devolver los productos EOL a un estado "como nuevo", ayudando a reducir la demanda de materias primas, el consumo energético y la generación de residuos. Así, la optimización de los procesos involucrados en este tipo de sistemas es una importante tarea que ataca directamente a los Objetivos de Desarrollo Sostenible ODS 9 (industria, innovación e infraestructura) y ODS 12 (producción y consumo responsables).

En concreto, este trabajo se centra en los sistemas de remanufactura en tres etapas (desensamblado, reprocesamiento y ensamblado), abreviado como 3T-RSSP (del inglés, *Three-Stage Remanufacturing System Scheduling Problem*), con aplicaciones en la industria automovilística, aerogeneradora o aeroespacial [7, 8]. El objetivo del 3T-RSSP es planificar la secuencia de procesamiento de un conjunto de productos, cada uno con varios componentes, en un sistema de remanufactura a través de estas tres etapas, tratando de optimizar una o varias métricas de eficiencia, como pueden ser la energía consumida, el tiempo de finalización de los trabajos (*makespan*) o la reducción en las emisiones de carbono. El 3T-RSSP es un problema que pertenece a la familia de problemas conocida como FSSP (*Flow Shop Scheduling Problem*), donde un conjunto de trabajos debe ser procesado secuencialmente a través de etapas ejecutadas por máquinas en serie [3]. En concreto, el 3T-RSSP considera la

existencia de múltiples máquinas en paralelo en al menos una de las etapas de procesamiento, como en la variante híbrida del FSSP [9]. Adicionalmente, el 3T-RSSP también comparte características con el UPMSP (*Unrelated Parallel Machine Scheduling Problem*), ya que las máquinas paralelas, aunque realizan las mismas tareas, no son iguales, por lo que el tiempo de procesado para un mismo producto puede variar en función de la máquina en la que se procese [1].

El 3T-RSSP es un problema *NP-Duro* que se ha trabajado desde enfoques tanto exactos como heurísticos. En la literatura se pueden encontrar diferentes variantes del problema en función de las características del sistema y objetivos a optimizar. Kim *et al.* proponen en [5] uno de los primeros modelos para una versión del 3T-RSSP con una única máquina en las etapas de desensamblado y ensamblado, pero manteniendo múltiples máquinas funcionando en paralelo en la etapa de reprocesamiento. En concreto, resuelven el problema mediante un algoritmo exacto y, adicionalmente, proponen tres heurísticas y una versión de la metaheurística *Iterated Greedy* para resolver instancias de tamaño entre 20 y 80 productos. Por otro lado, Wang *et al.* presentan en [10] la posibilidad de operar con varias máquinas en paralelo en las etapas de desensamblado y ensamblado. En particular, proponen un algoritmo genético para resolver instancias que incluyen entre 10 y 40 productos, con el objetivo de minimizar la energía consumida por el sistema. Más adelante, Wang propone en 2024 dos trabajos donde el objetivo es la minimización del *makespan*. En concreto, en el trabajo [11] propone un modelo exacto junto con 18 heurísticas para resolver instancias de tamaños entre 4 y 40 productos, mientras que en [12] añade otros cuatro modelos exactos para resolver instancias entre 4 y 200 productos.

En este trabajo se propone la aplicación de la metaheurística GRASP para resolver eficientemente el 3T-RSSP considerando el *makespan* como función objetivo. Para la fase constructiva se utiliza la variante clásica, donde en primer lugar se utiliza una función voraz para seleccionar de una lista los posibles candidatos a formar parte de la solución inicial, y en segundo lugar entra en juego la componente aleatoria. Para la fase de mejora se propone una búsqueda local que explora un vecindario extendido generado a partir de movimientos de intercambio e inserción. La mejor variante del algoritmo propuesto se compara con el estado del arte sobre un total de 34 instancias, obteniendo resultados muy competitivos y reduciendo el tiempo de cómputo en dos órdenes de magnitud.

El resto de este trabajo se estructura de la siguiente forma. En la Sección 2 se ofrece una descripción formal del problema 3T-RSSP. La Sección 3 presenta la propuesta algorítmica para el 3T-RSSP basada en la metaheurística GRASP. La Sección 4 está dedicada a la evaluación experimental de dicha propuesta frente al estado del arte. Finalmente, en la Sección 5, se resumen las principales conclusiones del trabajo y se proponen futuras líneas de investigación.

## 2  Descripción del problema

El 3T-RSSP bajo estudio considera $n$ productos EOL, cada uno con $r$ componentes, que llegan a un sistema de remanufactura compuesto por $d$ máquinas de desensamblado, $r$ líneas de reprocesamiento (dedicadas a cada uno de los $r$ componentes) y $e$ máquinas de ensamblado. Además, se conocen el tiempo de desensamblado de cada producto $p$ en cada máquina de desensamblado $m$, $TD_{pm}$, el tiempo de procesado de cada componente $r$ de cada producto $p$ en cada fase $f$ de cada línea de reprocesamiento $r$, $TR_{prf}$, y el tiempo de ensamblado de cada producto $p$ en cada máquina de ensamblado $m$, $TE_{pm}$.

La Figura 1 muestra un ejemplo de este tipo de sistemas, en el que 2 máquinas desensamblan los productos que van llegando al sistema en sus componentes a medida que estas están libres. A continuación, cada componente entra en una línea de reprocesamiento dedicada para ser sometido a los procesos correspondientes. En el ejemplo de la figura, se dispone de una primera línea que realiza tres procesos (por ejemplo, lijado, pintado y pulido), mientras que la segunda línea está dedicada a los componentes que únicamente requieran los dos procesos que dicha línea lleva a cabo. Una vez reprocesados todos los componentes de cada producto, se dispone de 2 líneas de ensamblado para devolver el producto a su nuevo estado tan pronto como alguna de las máquinas esté libre.

El objetivo del 3T-RSSP es minimizar el tiempo de remanufactura (*makespan*) de una serie de productos en un sistema con tres etapas como el aquí descrito. Dicho tiempo depende del orden en el que los productos que llegan al sistema se someten a los diferentes procesos en cada una de estas tres etapas. Denotando con $D_m$, $R_m$ y $E_m$, la secuencia ordenada de productos en la *m-ésima* máquina (o línea de reprocesamiento) de cada etapa, desensamblado, reprocesamiento y

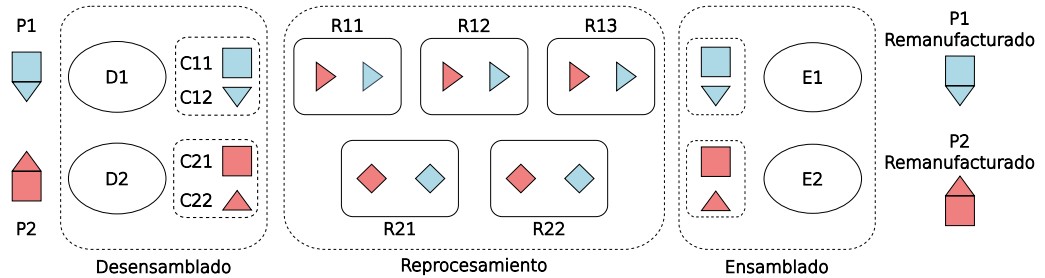

Figura 1: Ejemplo de sistema de manufactura con tres etapas.

ensamblado, respectivamente, una solución al problema se puede escribir como $\varphi = \{\pi_D, \pi_R, \pi_E\}$, donde $\pi_D = \{D_1, ..., D_d\}$, $\pi_R = \{R_1, ..., R_r\}$ y $\pi_E = \{E_1, ..., E_e\}$.

Para ilustrar mejor el problema, así como el cálculo de la función objetivo, supóngase que al sistema de la Figura 2 llegan 5 productos con 2 componentes cada uno. Una posible solución podría estar dada por las siguientes secuencias de desensamblado $D_1 = \{3, 5, 2\}$ y $D_2 = \{1, 4\}$, de reprocesamiento $R_1 = \{3, 5, 2, 1, 4\}$ y $R_2 = \{3, 2, 1, 5, 4\}$, y de ensamblado $E_1 = \{3, 2, 4\}$ y $E_2 = \{5, 1\}$. Dada esta solución, y conocidos los tiempos de procesamiento de cada uno de los productos y componentes en cada una de las etapas, el diagrama de Gantt de la Figura 2 muestra la evolución de las diferentes tareas/procesos a lo largo del tiempo cuando el sistema funciona de acuerdo a la solución anterior.

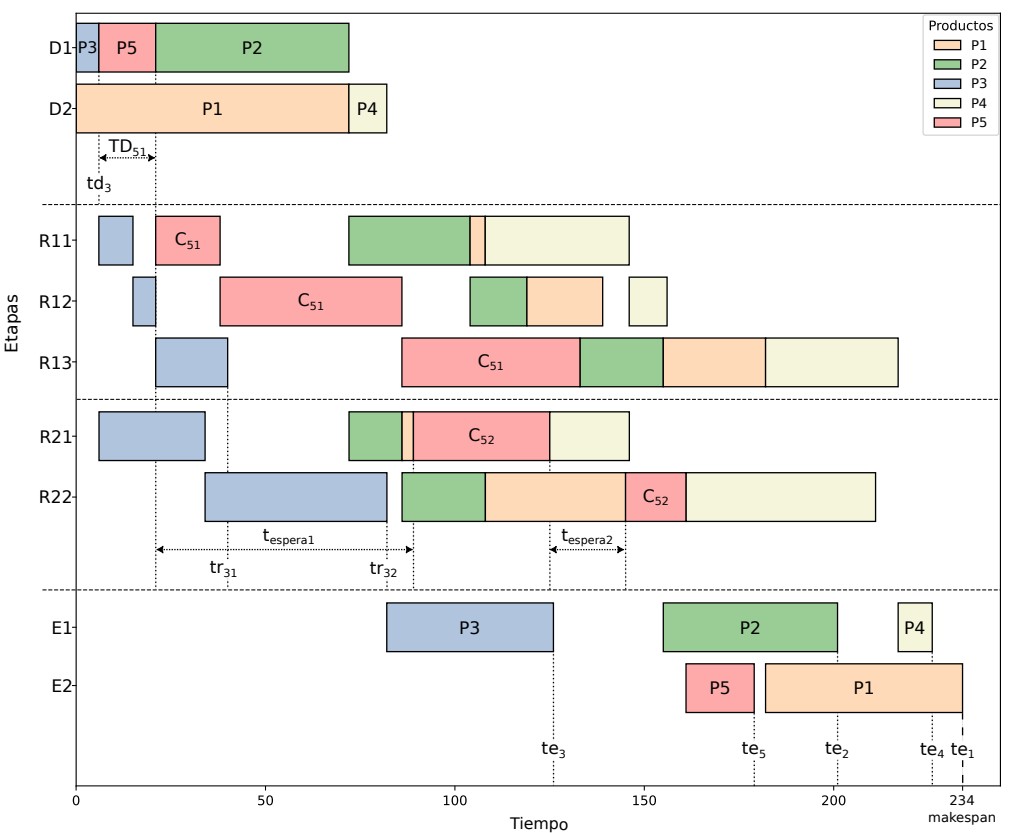

Figura 2: Diagrama de Gantt de una solución con cinco productos.

Sean $td_p$, $tr_{pr}$, $te_p$ los instantes de tiempo (desde el comienzo del proceso) en los que producto $p$ finaliza su desensamblado, reprocesamiento en la línea $r$ asignada al componente correspondiente, y

ensamblado, respectivamente. Siguiendo con la solución de la Figura 2 y tomando como ejemplo el producto $P5$, el tiempo de remanufactura del dicho producto en el sistema es $te_5$. Para llegar a dicho tiempo, el producto $P5$ comienza su desensamblado en la primera máquina que, dado que está ocupada por el producto $P3$, no puede comenzar hasta que dicha máquina esté libre. Así, el producto $P5$ finaliza su desensamblado en el instante $td_5 = td_3 + TD_{51}$. A continuación, los dos componentes de $P5$ ($C_{51}$ y $C_{52}$) están disponibles para ser reprocesados en las líneas de reprocesamiento de la segunda etapa. Como puede verse en la Figura, el componente $C_{51}$ es sometido a los tres procesos de la primera línea sin esperas, finalizando su reprocesado en el instante $tr_{51} = td_5 + TR_{511} + TR_{512} + TR_{513}$. Por otro lado, el componente $C_{52}$ debe esperar a que la segunda línea esté libre, haciendo el instante de tiempo en el que finalice su reprocesado sea $tr_{52} = t_{d5} + t_{espera} + TR_{521} + TR_{522}$, donde $t_{espera}$, que en el ejemplo es la suma entre $t_{espera1}$ y $t_{espera2}$, indica el tiempo de espera entre $t_{d5}$ y el comienzo de su procesado en la segunda línea como consecuencia del bloqueo de la misma por el procesado de otros componentes. Una vez los dos componentes han sido reprocesados, ya puede comenzar el reensamblado del producto $P5$ en la tercera etapa. En este caso, dado que este ensamblado tiene lugar en la segunda máquina y esta está libre, la remanufactura del producto $P5$ finaliza en el instante $te_5 = max(tr_{51}, tr_{52}) + TE_5$. Nótese que, al igual que con el componente de la segunda etapa, si la máquina de ensamblado hubiese estado ocupada, al cálculo de $te_5$ habría que sumarle un tiempo de espera. Una vez calculados el tiempo de ensamblado de cada uno de los productos $p \in \{1, ..., n\}$, el *makespan* del sistema bajo la solución $\varphi$ se calcula como:

$$\mathcal{F}(\varphi) = \max_{p \in \{1,...,n\}} \{te_p\} \tag{1}$$

Por tanto, el objetivo del 3T-RSSP de minimizar el makespan, puede escribirse formalmente como se muestra en la Ecuación (2), donde $\Phi$ es el conjunto de todas las soluciones factibles.

$$\varphi^* = \arg\min_{\varphi \in \Phi} \mathcal{F}(\varphi) \tag{2}$$

## 3 Propuesta algorítmica

En este trabajo se propone el uso de un método GRASP (del inglés, *Greedy Randomized Adaptive Search Procedure*) para resolver el problema 3T-RSSP. GRASP es una metaheurística creada en 1989 por Feo y Resende [2]. Tal y como se muestra en el Algoritmo 1, es un procedimiento multiarranque en el que cada iteración tiene dos fases diferenciadas: una primera fase que construye una solución inicial siguiendo un procedimiento voraz-aleatorizado que depende del parámetro $\alpha$ (línea 3), seguida de una segunda fase que trata de mejorar la calidad de la solución obtenida en la primera usando un método de búsqueda local (línea 4). El procedimiento finaliza devolviendo en la línea 8 la mejor solución obtenida en las $t_{max}$ iteraciones ejecutadas, cuya actualización se lleva a cabo en las líneas 5 a 7.

---

**Algoritmo 1:** GRASP $(t_{max}, \alpha)$

**1** $\mathcal{F}^* \leftarrow \infty$
**2** **for** $t = 1$ **to** $t_{max}$ **do**
**3** $\quad$ $\varphi \leftarrow$ `Constructivo`$(\alpha)$
**4** $\quad$ $\varphi' \leftarrow$ `BusquedaLocal`$(\varphi)$
**5** $\quad$ **if** $\mathcal{F}(\varphi') < \mathcal{F}^*$ **then**
**6** $\quad\quad$ $\mathcal{F}^* \leftarrow \mathcal{F}(\varphi')$
**7** $\quad\quad$ $\varphi^\star \leftarrow \varphi'$

**8** **return** $\varphi^\star$

---

### 3.1 Método constructivo

El propósito de la fase constructiva es generar una solución inicial con cierta calidad. Se trata de un método iterativo en el que, partiendo de una solución vacía, se va añadiendo un nuevo elemento en cada iteración. Esta fase suele estar guiada por una función de selección voraz que, junto con un parámetro $\alpha$, ayuda al método a elegir los elementos más prometedores para ser incluidos en la solución parcial. En esta propuesta, los productos se van asignando a las secuencias de procesamiento de las diferentes máquinas en cada etapa de uno en uno, utilizando el *makespan* de la solución parcial como función voraz.

El Algoritmo 2, muestra el pseudocódigo de la fase de construcción, donde, partiendo de una solución vacía, las funciones Desensamblado, Reprocesamiento y Ensamblado van añadiendo a la solución parcial las secuencias de desensamblado, reprocesamiento y ensamblado, respectivamente, siendo detallado en el Algoritmo 3 para la etapa de desensamblado. El procedimiento es similar para las etapas de reprocesamiento y ensamblado, con la salvedad de que en la etapa de reprocesamiento la línea 5 del Algoritmo 3 solo incluye el tipo de producto en lugar de los pares producto-máquina.

---

**Algoritmo 2:** Constructivo$(\alpha)$

---

1 $\varphi \leftarrow \emptyset$

2 $\varphi \leftarrow$ Desensamblado$(\varphi, \alpha)$

3 $\varphi \leftarrow$ Reprocesamiento$(\varphi, \alpha)$

4 $\varphi \leftarrow$ Ensamblado$(\varphi, \alpha)$

5 **return** $\varphi$

---

El Algoritmo 3 comienza inicializando la secuencia de cada una de las máquinas de desensamblado a un conjunto vacío en las líneas 1 y 2. A continuación, tras inicializar la lista de candidatos $CL$ con el conjunto de todos los productos, se entra en un bucle (líneas 4 a 12) que construye la secuencia inicial añadiendo un elemento a la solución parcial en cada iteración. Para ello, dicho bucle comienza generando la lista de candidatos extendida *ECL* que incluye todas las posibles combinaciones de producto $p \in CL$ y máquina de desensamblado $m \in \{1, \dots, d\}$ (línea 5). A continuación, se obtienen los valores mínimo y máximo de la función voraz (líneas 6 y 7), denotada como $\mathcal{G}(\varphi, p, m)$, que calcula el valor de la función objetivo si a la solución $\varphi$ se le asigna el producto $p$ en la máquina $m$. Con estos valores y el parámetro $\alpha$ se genera la lista de candidatos reducida *RCL* en la línea 8. Finalmente, tras seleccionar aleatoriamente un elemento de la *RCL* en la línea 9, este es utilizado en las líneas 10 y 11 para actualizar la solución parcial. La línea 12 elimina de la *CL* el producto $p$ incluido en la solución, y el bucle continúa hasta que no queden elementos en dicha lista, devolviendo la solución construida.

### 3.2 Búsqueda Local

La búsqueda local propuesta en este trabajo para la fase de mejora se basa en la exploración de un vecindario extendido generado a partir de dos movimientos diferentes: *inserción* e *intercambio*. Dichos movimientos se aplican de manera análoga dentro de cada una de las tres etapas. Por simplicidad, a continuación solo se explican estos movimientos en la etapa de desensamblado, siendo aplicables de manera análoga en las otras dos etapas.

El movimiento de *inserción* consiste en eliminar un producto $p$ ubicado en la posición $i$ de la secuencia de desensamblado de la máquina $m$, $D_m$, para a continuación insertarlo en otra posición diferente $j \neq i$ de la misma máquina $m$, o en cualquiera de las posiciones de la secuencia de desensamblado de otra máquina $m' \neq m$, $D_{m'}$. Así, siguiendo con la solución de la Figura 2, donde la secuencia de ensamblado es $\pi_D = \{\{3, 5, 2\}, \{1, 4\}\}$, ejemplos de posibles movimientos de inserción en la etapa de desensamblado son $\pi'_D = \{\{5, 3, 2\}, \{1, 4\}\}$, donde el producto $P3$ ha modificado su posición en la primera máquina, o $\pi'_D = \{\{5, 2\}, \{1, 4, 3\}\}$, donde el producto $P3$ se coloca en la última posición de la segunda máquina. Como se ha comentado más arriba, este movimiento también es aplicable a las etapas de ensamblado o reprocesamiento, con la única excepción de que en la etapa de reprocesamiento las inserciones solo pueden tener lugar dentro de la misma línea. Siguiendo con el ejemplo, donde la secuencia de reprocesamiento es $\pi_R = \{\{3, 5, 2, 1, 4\}, \{3, 2, 1, 5, 4\}\}$, un posible movimiento es $\pi'_R = \{\{5, 2, 1, 3, 4\}, \{3, 2, 1, 5, 4\}\}$, donde el producto $P3$ ha modificado

**Algoritmo 3:** Desensamblado$(\varphi, \alpha)$

---

**1 for** $m = 1$ **to** $d$ **do**

**2**     $D_m \leftarrow \emptyset$

**3** $CL \leftarrow \{1, \ldots, n\}$

**4 while** $|CL| > 0$ **do**

**5**     $ECL \leftarrow \{(p, m) \mid p \in CL \wedge m \in \{1, \ldots, d\}\}$

**6**     $\mathcal{G}_{min} \leftarrow \min\limits_{(p,m) \in ECL} \mathcal{G}(\varphi, p, m)$

**7**     $\mathcal{G}_{max} \leftarrow \max\limits_{(p,m) \in ECL} \mathcal{G}(\varphi, p, m)$

**8**     $RCL \leftarrow \{(p, m) \in ECL : \mathcal{G}(\varphi, p, m) \leq \mathcal{G}_{max} - \alpha \cdot (\mathcal{G}_{max} - \mathcal{G}_{min})\}$

**9**     $(p, m) \leftarrow \texttt{SeleccionarAleatorio}(RCL)$

**10**     $D_m \leftarrow D_m \cup \{p\}$

**11**     $\varphi \leftarrow \texttt{Actualizar}(\varphi, D_m)$

**12**     $CL \leftarrow CL \setminus \{p\}$

**13 return** $\varphi$

---

su posición en la secuencia de la primera línea de la etapa de reprocesamiento (nótese que dicho producto no puede ser insertado en $R_2$).

El movimiento de *intercambio* consiste en intercambiar un producto $p$ ubicado en la posición $i$ de la secuencia de desensamblado de la máquina $m$, $D_m$, por otro producto $p'$ en otra posición diferente $j \neq i$ de la misma máquina $m$, o en cualquiera de las posiciones de la secuencia de desensamblado de otra máquina $m' \neq m$, $D_{m'}$. Así, siguiendo con la solución de la Figura 2, donde la secuencia de desensamblado es $\pi_D = \{\{3, 5, 2\}, \{1, 4\}\}$, ejemplos de posibles movimientos de intercambio en la etapa de ensamblado son $\pi'_D = \{\{2, 5, 3\}, \{1, 4\}\}$ donde los productos $P2$ y $P3$ han intercambiado posiciones en la primera máquina, o $\pi'_D = \{\{3, 5, 1\}, \{2, 4\}\}$ donde los productos $P1$ y $P2$ han intercambiado posición y máquina. Al igual que con el movimiento de *inserción*, el movimiento de *intercambio* es aplicable a las etapas de desensamblado y reprocesamiento, exceptuando que en la etapa de reprocesamiento el movimiento solo puede tener lugar dentro de la misma línea.

El Algoritmo 4 muestra el pseudocódigo de la búsqueda local propuesta para el 3T-RSSP, que se basa en la exploración de los vecindarios extendidos generados con ambos movimientos, (*inserción* e *intercambio*), de manera secuencial en cada una de las tres etapas (desensamblado, reprocesamiento y ensamblado). Tras inicializar la variable *mejora* en la línea 1, la búsqueda local entra en un bucle (líneas 2 a 10) que explora los vecindarios extendidos de cada una de las tres etapas en las líneas 4, 5 y 6. A continuación, la función `ObtenerMejor` devuelve en la línea 7 la mejor solución de las tres obtenidas, que se utiliza para actualizar la solución actual en caso de mejora (líneas 8 a 10). El bucle continúa hasta que no es posible encontrar ninguna mejora en los vecindarios explorados.

## 4   Resultados experimentales

El GRASP propuesto en este trabajo para el 3T-RSSP se ha evaluado frente al estado del arte sobre el conjunto de 34 instancias propuesto en [11]. Estas instancias están etiquetadas como `Pp_Cr_DdAe`, donde $p$ hace referencia al número de productos, $r$ hace referencia al número de componentes o líneas de reprocesamiento, $d$ al número de máquinas de desensamblado y $e$ el número de máquinas de ensamblado. En concreto, las instancias de este conjunto tienen entre 4 y 40 productos, entre 1 y 6 máquinas de desensamblado, entre 2 y 5 líneas de reprocesamiento (componentes), y entre 1 y 3 máquinas de ensamblado. Toda la experimentación ha sido realizada en una máquina con procesador AMD EPYC 7643 16-core virtual CPU con 8 GB de RAM, usando Java 21.

### 4.1   Ajuste de parámetros

Como se ha comentado en la Sección 3.1, el algoritmo aquí propuesto depende del parámetro $\alpha$ que regula el equilibrio entre una selección totalmente aleatoria ($\alpha = 0$) o voraz ($\alpha = 1$) de los

---
**Algoritmo 4:** BusquedaLocal($\varphi$)
---
**1**   $mejora \leftarrow$ true

**2**   **while** $mejora$ **do**

**3**      $mejora \leftarrow$ false

**4**      $\varphi_D \leftarrow$ ExplorarD($\varphi$)

**5**      $\varphi_R \leftarrow$ ExplorarR($\varphi$)

**6**      $\varphi_E \leftarrow$ ExplorarE($\varphi$)

**7**      $\varphi' \leftarrow$ ObtenerMejor($\varphi_D, \varphi_R, \varphi_E$)

**8**      **if** $\mathcal{F}(\varphi') < \mathcal{F}(\varphi)$ **then**

**9**         $\varphi \leftarrow \varphi'$

**10**        $mejora \leftarrow$ true

**11**   **return** $\varphi$
---

elementos a incluir en la solución inicial en la fase de construcción. Por tanto, para encontrar la mejor configuración del algoritmo, se ha realizado un experimento en el que se compara el rendimiento del mismo bajo diferentes valores del parámetro $\alpha$. En concreto, se han examinado los valores $\alpha = \{0, 0; 0, 25; 0, 5; 0, 75; 1, 0\}$.

La Tabla 1 muestra el rendimiento del algoritmo para los diferentes valores de $\alpha$ estudiados utilizando como criterio de parada $t_{max} = 100$. Para cada valor de $\alpha$ se muestra el promedio de diferentes métricas sobre los resultados obtenidos en las 34 instancias bajo estudio, destacando en negrita el mejor valor de cada métrica. En particular, la primera columna muestra el coste de la función objetivo (Coste); la segunda, la desviación porcentual de dicho coste respecto al mejor valor encontrado en este experimento (Desv. (%)); la tercera, el número de instancias en que la configuración asociada a cada valor de $\alpha$ obtiene la mejor solución encontrada (#Best); y finalmente, la cuarta columna muestra el tiempo de ejecución en segundos (Tiempo (s)).

Tabla 1: Rendimiento del algoritmo para diferentes valores de $\alpha$.

| $\alpha$ | Coste | Desv. (%) | #Best | Tiempo (s) |
|---|---|---|---|---|
| 0,00 | **453,88** | **0,28** | 26 | 9,24 |
| **0,25** | 454,06 | 0,71 | **28** | 6,04 |
| 0,50 | 454,03 | 0,74 | 26 | 3,34 |
| 0,75 | 455,29 | 1,12 | 19 | 1,33 |
| 1,00 | 459,53 | 2,61 | 10 | **1,23** |

Como se observa en la tabla, a medida que se incrementa la voracidad en la construcción de la solución inicial, aumentando el valor de $\alpha$, el tiempo de ejecución del algoritmo se reduce. Esto es debido a que a mayor calidad de la solución generada por el método constructivo, menor es el recorrido de la búsqueda local antes de caer en un mínimo local. Por otro lado, se observa que incluyendo algo de aleatoriedad en la construcción de las soluciones iniciales, reduciendo el valor de $\alpha$, la búsqueda local es capaz de avanzar más, incrementando la calidad de las soluciones finales. En particular, el mejor valor del coste y desviación lo ofrece la versión con $\alpha = 0, 0$, mientras que la versión con $\alpha = 0, 25$ obtiene la mejor solución en 28 de las 34 instancias. Finalmente, aún con un coste promedio ligeramente inferior, dado el mayor número de instancias donde obtiene la mejor solución y la reducción del tiempo de ejecución en un 34,63% respecto a la versión con $\alpha = 0, 0$, se selecciona la versión con $\alpha = 0, 25$ como la mejor configuración del GRASP aquí propuesto de cara a la comparativa con el estado del arte.

## 4.2   Comparativa con estado del arte

Una vez ajustado el parámetro $\alpha$, en esta sección se compara la mejor versión del algoritmo GRASP propuesto para el 3T-RSSP ($\alpha = 0, 25$) frente al estado del arte. En particular, el trabajo [11] propone 18 heurísticas constructivas y el uso de CPLEX para resolver el modelo de programación lineal entera mixta (MILP) propuesto, limitando el tiempo de ejecución a 3600 segundos. Adicionalmente, el

trabajo [12] propone cuatro modelos exactos (Modelo 1 a Modelo 4), limitando a 600 segundos la ejecución en las instancias pequeñas y 3600 segundos en las grandes. Cabe destacar que en el primer trabajo ([11]) no se proporcionan las características de la CPU en la que se han ejecutado los experimentos, mientras que el segundo ([12]) utiliza un Intel i5-12500H que, de acuerdo a `http://www.cpubenchmark.net/`, tiene un rendimiento en ejecución *single-thread* de un 21.8% superior a la CPU utilizada en el trabajo aquí propuesto.

La Tabla 2 muestra la comparativa frente al estado del arte del GRASP propuesto para dos valores diferentes de $t_{max}$, utilizando las mismas métricas de la Tabla 1. Una primera versión que utiliza un valor de $t_{max} = 100$, denotada como GRASP-100, y una segunda versión con $t_{max} = 1000$, denotada como GRASP-1000. Estas dos versiones del algoritmo buscan un equilibrio entre calidad de la solución y tiempo de ejecución. En esta tabla es relevante destacar que los modelos etiquetados con el símbolo $\star$ no llegan a resolver las 34 instancias en el límite de tiempo establecido. En particular, el modelo MILP presenta problemas de memoria en una de las instancias, por lo que sus resultados corresponden a 33 de las 34 instancias. Por otro lado, el Modelo 3 tan solo obtiene un resultado factible para 3 de las 34 instancias. Asimismo, el Modelo 4 resuelve de manera factible 32 instancias. Es por esto que los valores agregados de las métricas para los modelos etiquetados con el símbolo $\star$ no han sido tomados en cuenta en la comparativa final.

Tabla 2: Comparativa frente al estado del arte. El símbolo $\star$ indica que no se han resuelto las 34 instancias en el límite de tiempo establecido.

| Algoritmo | Coste | Desv. (%) | #Best | Tiempo (s) |
|---|---|---|---|---|
| LAPT-F | 465,03 | 4,56 | 13 | **0,20** |
| LAPT-H | 488,74 | 11,64 | 1 | **0,20** |
| LAPT-L | 589,47 | 43,26 | 0 | **0,20** |
| LTPT-F | 464,00 | 4,29 | 12 | **0,20** |
| LTPT-H | 485,44 | 10,85 | 2 | **0,20** |
| LTPT-L | 585,97 | 40,99 | 0 | **0,20** |
| LTRT-F | 462,47 | 4,11 | 12 | **0,20** |
| LTRT-H | 493,62 | 13,13 | 4 | **0,20** |
| LTRT-L | 607,29 | 43,69 | 0 | **0,20** |
| SAPT-F | 463,68 | 4,71 | 9 | **0,20** |
| SAPT-H | 496,00 | 12,82 | 3 | **0,20** |
| SAPT-L | 602,29 | 45,23 | 0 | **0,20** |
| STPT-F | 465,12 | 4,99 | 11 | **0,20** |
| STPT-H | 495,12 | 13,31 | 3 | **0,20** |
| STPT-L | 599,26 | 45,62 | 0 | **0,20** |
| STRT-F | 465,85 | 5,07 | 9 | **0,20** |
| STRT-H | 504,82 | 14,36 | 2 | **0,20** |
| STRT-L | 607,12 | 44,70 | 0 | **0,20** |
| MILP$\star$ | 441,70 | 1,55 | 26 | 1602,64 |
| Modelo 1 | 458,47 | **0,53** | **30** | 782,29 |
| Modelo 2 | 521,65 | 5,14 | 27 | 771,22 |
| Modelo 3$\star$ | 279,00 | 0,00 | 3 | 1473,82 |
| Modelo 4$\star$ | 408,69 | 1,48 | 25 | 1149,81 |
| GRASP-100 | 454,06 | 0,91 | 26 | 6,04 |
| GRASP-1000 | **453,26** | 0,68 | 28 | 58,62 |

Se puede observar que las 18 heurísticas constructivas generan soluciones empleando muy poco tiempo, pero en general tienen poca calidad. Los métodos exactos mejoran claramente los resultados de las heurísticas, especialmente el Modelo 1, que obtiene el mejor resultado en 30 instancias y la menor desviación, aunque empleando tiempos considerablemente elevados. Cabe destacar que los resultados obtenidos por el método GRASP propuesto en este trabajo alcanzan un buen compromiso entre calidad y tiempo de ejecución, llegando a obtener el mejor valor promedio para la función de coste y, en el caso de la versión GRASP-1000, el mejor valor posible en 28 instancias empleando menos del 7,5% del tiempo utilizado por el modelo matemático.

Finalmente, la Tabla 3 muestra el detalle de los resultados del algoritmo GRASP propuesto en este trabajo considerando los dos límites de 100 y 1000 iteraciones para cada una de las instancias. Se han destacado en cursiva los mejores valores encontrados y en negrita los óptimos.

Tabla 3: Detalle de los resultados obtenidos por GRASP-100 y GRASP-1000 en las 34 instancias.

| | GRASP-100 | | GRASP-1000 | |
|---|---|---|---|---|
| Instancia | Coste | Tiempo (s) | Coste | Tiempo (s) |
| P4_C2_D1A2 | *295* | 0,02 | *295* | 0,11 |
| P4_C2_D2A1 | **267** | 0,01 | **267** | 0,10 |
| P4_C2_D2A2 | **197** | 0,01 | **197** | 0,08 |
| P4_C2_D4A2 | **197** | 0,01 | **197** | 0,09 |
| P4_C3_D1A2 | **180** | 0,01 | **180** | 0,11 |
| P4_C3_D2A1 | **238** | 0,01 | **238** | 0,14 |
| P4_C3_D2A2 | **153** | 0,02 | **153** | 0,18 |
| P4_C3_D4A2 | **150** | 0,02 | **150** | 0,20 |
| P5_C2_D1A2 | *314* | 0,01 | *314* | 0,08 |
| P5_C2_D2A1 | **301** | 0,01 | **301** | 0,16 |
| P5_C2_D2A2 | **235** | 0,02 | **235** | 0,18 |
| P5_C2_D4A2 | **216** | 0,02 | **216** | 0,25 |
| P5_C3_D1A2 | **273** | 0,03 | **273** | 0,21 |
| P5_C3_D2A1 | **279** | 0,08 | **279** | 0,31 |
| P5_C3_D2A2 | **234** | 0,04 | **234** | 0,33 |
| P5_C3_D4A2 | **244** | 0,04 | **244** | 0,33 |
| P8_C2_D1A2 | *412* | 0,06 | *412* | 0,47 |
| P8_C2_D2A1 | **459** | 0,07 | **459** | 0,70 |
| P8_C2_D2A2 | 242 | 0,10 | *229* | 1,03 |
| P8_C2_D4A2 | 232 | 0,10 | *229* | 1,00 |
| P8_C3_D1A2 | *405* | 0,17 | *405* | 1,61 |
| P8_C3_D2A1 | **398** | 0,17 | **398** | 1,53 |
| P8_C3_D2A2 | **309** | 0,13 | **309** | 1,40 |
| P8_C3_D4A2 | **313** | 0,14 | **313** | 1,31 |
| P10_C2_D2A2 | *376* | 0,18 | *376* | 1,82 |
| P10_C3_D4A2 | *377* | 0,25 | *377* | 2,56 |
| P10_C5_D6A3 | *377* | 0,60 | *377* | 5,29 |
| P20_C3_D6A3 | *710* | 1,88 | *710* | 17,83 |
| P20_C5_D4A2 | *718* | 6,58 | *718* | 63,56 |
| P20_C5_D6A3 | *718* | 4,84 | *718* | 46,23 |
| P40_C3_D4A2 | *1402* | 33,17 | *1402* | 319,93 |
| P40_C3_D6A3 | *1402* | 20,68 | *1402* | 197,05 |
| P40_C5_D4A2 | 1413 | 84,11 | *1402* | 822,04 |
| P40_C5_D6A3 | *1402* | 51,91 | *1402* | 504,93 |
| **Promedio** | 454,06 | 6,04 | 453,26 | 58,62 |

# 5 Conclusiones y trabajo futuro

En este trabajo se ha abordado el problema de la optimización de sistemas de remanufactura en tres etapas mediante el uso de la metaheurística GRASP. El objetivo principal ha sido minimizar el *makespan* del sistema, contribuyendo así a la eficiencia y sostenibilidad de los procesos de remanufactura, en línea con los Objetivos de Desarrollo Sostenible 9 y 12.

Los resultados experimentales, obtenidos para 34 instancias, demuestran que el método GRASP propuesto ofrece un buen equilibrio entre calidad de solución y tiempo de cómputo. En la comparativa con el estado del arte, el método propuesto ha obtenido el mejor coste promedio empleando menos de un 7,5% del tiempo usado por el mejor método exacto.

Como trabajo futuro se planea mejorar el algoritmo desarrollado por medio de diferentes estrategias constructivas y de mejora. También se está estudiando mejorar la eficiencia del algoritmo con el objetivo de resolver instancias de mayor tamaño. Adicionalmente, se plantea la utilización de otras metaheurísticas o el uso de otras métricas de eficiencia como función objetivo.

## Agradecimientos y declaración de financiación

Este trabajo forma parte de los proyectos RED2022-134480-T y PID2021-125709OA-C22 financiados por el Ministerio de Ciencia e Innovación (MCIN/AEI/10.13039/501100011033) y FEDER, *una manera de hacer Europa*, así como del proyecto TEC-2024/COM-404 financiado por la Comunidad Autónoma de Madrid.

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
