# OpenReview forum: "Un algoritmo GRASP para el sistema de remanufactura en tres etapas"
_MAEB/2025/Congreso — MAEB 2025_

### Official Review · Reviewer_w8W7 · 2025-03-17
**El trabajo presenta una solución basada en la metaheurística GRASP para optimizar sistemas de manufactura en tres etapas.**

**Rating:** 4
**Confidence:** 4

**Review:**

El trabajo presenta una solución basada en la metaheurística GRASP para optimizar sistemas de manufactura en tres etapas, mostrando buenos resultados en términos de calidad y eficiencia. Sin embargo, carece de una sección clara que destaque sus aportaciones novedosas y una comparación explícita con otros enfoques, sobre todos los empleados con la experimentación que está muy relacionada, por lo que el impacto no se puede valorar de forma adecuada. Estos métodos constructivos seguro que se pueden utilizar de forma aleatorizada para usarse como paso 1 de  GRASP. Se recomienda incluir una sección sobre contribuciones específicas del método, una discusión sobre por qué GRASP es más adecuado que otras metaheurísticas y un análisis más profundo de los resultados experimentales para interpretar mejor su rendimiento.

---

### Official Review · Reviewer_jSVw · 2025-03-17
**Un algoritmo GRASP para el sistema de remanufactura en tres etapas**

**Rating:** 5
**Confidence:** 5

**Review:**

En este trabajo se resuelve el problema de optimización de sistemas de remanufactura en tres etapas (3T-RSSP) con el objetivo de minimizar el tiempo de finalización de los sistemas de trabajo (makespan). Para resolver el problema se utiliza una metaheurística GRASP y se resuelven 34 instancias. GRASP ofrece un equilibrio entre la calidad de la solución y el tiempo de cómputo. El artículo es interesante y está bien escrito, además, la comparativa es adecuada.
A continuación, se detallan algunas revisiones menores que se proponen a los autores:
- En lugar de poner “Kim y colaboradores” se propone que se cambie por “Kim et al.” «et al.» debe escribirse en cursiva. Revisar similares como “Wang y colaboradores”.
- Línea 76 página 2.  Revisar el subíndice de TR_prf porque aparece con un espacio.
-En la Figura 2 aparecen tiempo de espera: t_espera1 y t_espera2 pero no se indica la definición de t_espera1 y t_espera2. Hasta la línea 111 de la página 4 no está definido y aparece como t_espera y creo que sería adecuado definir qué es 1 y 2. En términos generales definiría t_esperaN donde habría que decir qué es N.
- En la línea 99 de la página 3 se define td_p, tr_pr, te_p, pero en la Figura 2 td_p, aparece con dos subíndices, tr_pr aparece con tres subíndices y te_p aparece con dos subíndices. Corregir.
- En la Sección de Ajuste de parámetros usted pone α = {0, 0,25, 0,50, 0,75,1}. En esa tabla a me faltaría alguna medida de variabilidad de las soluciones, tipo desviación típica o coeficiente de variación de Pearson ya que usted está dando valores medios y me gustaría saber cómo de representativos son esos costes medios.
- De nuevo, en la Sección de Ajuste de parámetros sería interesante ver qué ocurre considerando α uniforme y α random. Es decir, si ejecuta el algoritmo 100 veces, α uniforme serían diferentes valores de α entre [0,1] dando un salto de 1/100, por ejemplo. De forma similar, α random considera que en cada ejecución del algoritmo se toma un α seleccionado aleatoriamente en el intervalo [0,1]. Esto suele dar buenos resultados y no tiene la necesidad de realizar el ajuste de los parámetros y quedarse con un único valor de α, obteniéndose soluciones a la vez buenas y diversas.
- Línea 222 página 7 dice “a mayor calidad de la solución generada por el algoritmo, menor es el recorrido de la búsqueda local” sin embargo, me llama la atención que el tiempo de cómputo sea mayor. Entendí que la Tabla 1 ya lleva acoplada la búsqueda local, en cuyo caso es extraño que el algoritmo tarde más si la solución de partida en buena porque efectivamente tardaría muy poco en quedarse atrapado en un óptimo local. ¿Sabría decir cuánto tarda la construcción y cuánto tarda la búsqueda local?  Quizás es interesante mostrar cuánto tarda su algoritmo en la construcción y cuánto tarda en la búsqueda local. También me parecería interesante ver cuánto es capaz de mejorar la búsqueda local. Se me ocurre añadir dos columnas en esa tabla donde se incluyan tanto el coste como el tiempo de la fase de construcción.
- Tabla 2 indica que “el símbolo *” es que no sean resuelto las 34 instancias en el límite de tiempo establecido” y por tanto la comparativa no es justa para usted porque el coste obtenido por los modelos 3, 4 y MILP son engañosamente menores. Yo resaltaría eso porque puede resultar confuso.
- Por último, usted tarda con 1000 ejecuciones del algoritmo GRASP 58,62 segundos y el Modelo 1 tarda 782,29 segundos. ¿Ha comprobado que ocurre si usted aumenta considerablemente el número de ejecuciones del algoritmo GRASP hasta un tiempo similar al exacto, es decir, 782,29 segundos? Tal vez usted logre encontrar más de 30 mejores soluciones sin necesidad de alcanzar ese tiempo. Esto es simplemente una curiosidad y no hay necesidad de incluirlo ya que el rendimiento de su algoritmo es bastante bueno.

---

### Official Review · Reviewer_qv7u · 2025-03-19
**Metaheurístico para resolver el problema del Flow Shop Scheduling problem en un sistema de remanufactura de tres etapas**

**Rating:** 4
**Confidence:** 5

**Review:**

En este paper se propone el metaheurístico GRASP para resolver el problema del Flow Shop Scheduling problem cuyo objetivo es minimizar el makespan. Los autores comparan con 4 métodos exactos y 18 heurísticas constructivas siendo su método competitivo.

Científicamente no aporta ninguna novedad ya que el método es conocido y se ha aplicado previamente a este problema [1], sin embargo es interesante el enfoque de 3T-RSSP que le han dado. Han comparado con muchos métodos del estado del arte (la mayoría heurísticas) aunque hecho de menos que comparen con otras metaheurísticas como algoritmos genéticos, PSO...

El pseudocódigo del Algoritmo 2 sobra, no aporta información de interés.



[1] González-Neira, E. M., & Montoya-Torres, J. R. (2017). A GRASP meta-heuristic for the hybrid flowshop scheduling problem. Journal of Decision systems, 26(3), 294-306.

---

### Decision · Program_Chairs · 2025-03-20

Accept